# The Development of CDC25A-Derived Phosphoseryl Peptides That Bind 14-3-3ε with High Affinities

**DOI:** 10.3390/ijms25094918

**Published:** 2024-04-30

**Authors:** Seraphine Kamayirese, Sibaprasad Maity, Laura A. Hansen, Sándor Lovas

**Affiliations:** Department of Biomedical Sciences, Creighton University, Omaha, NE 68178, USA

**Keywords:** 14-3-3, phosphopeptides, inhibitors, molecular dynamics simulations, biophysical methods, cutaneous squamous cell carcinoma

## Abstract

Overexpression of the 14-3-3ε protein is associated with suppression of apoptosis in cutaneous squamous cell carcinoma (cSCC). This antiapoptotic activity of 14-3-3ε is dependent on its binding to CDC25A; thus, inhibiting 14-3-3ε – CDC25A interaction is an attractive therapeutic approach to promote apoptosis in cSCC. In this regard, designing peptide inhibitors of 14-3-3ε – CDC25A interactions is of great interest. This work reports the rational design of peptide analogs of pS, a CDC25A-derived peptide that has been shown to inhibit 14-3-3ε–CDC25A interaction and promote apoptosis in cSCC with micromolar IC_50_. We designed new peptide analogs in silico by shortening the parent pS peptide from 14 to 9 amino acid residues; then, based on binding motifs of 14-3-3 proteins, we introduced modifications in the pS(174–182) peptide. We studied the binding of the peptides using conventional molecular dynamics (MD) and steered MD simulations, as well as biophysical methods. Our results showed that shortening the pS peptide from 14 to 9 amino acids reduced the affinity of the peptide. However, substituting Gln^176^ with either Phe or Tyr amino acids rescued the binding of the peptide. The optimized peptides obtained in this work can be candidates for inhibition of 14-3-3ε – CDC25A interactions in cSCC.

## 1. Introduction

Various biological functions have been attributed to 14-3-3 proteins due to their interactions with a multitude of binding partners. The 14-3-3 proteins bind and regulate the activities of proteins, including protein kinase C [1,2], Raf-1 [3,4], Bcl [5], and CDC25A phosphatase [6]. Because of the large and diverse number of proteins that bind 14-3-3, this family of proteins plays various roles in key cellular processes, including cell proliferation, signal transduction [7], cell cycle control [8,9,10], and apoptosis [8].

14-3-3 proteins interact with their binding partners through a binding motif containing phosphoseryl (pSer) or phosphothreonyl (pThr) amino acid residue [6,11,12,13,14]. In a screening of a phosphopeptide-oriented library, Yaffe [11] identified two optimal binding motifs for 14-3-3 proteins, namely RSXpSXP and RXY/FXpSXP; pS denotes pSer, and X denotes any amino acid residue. Furthermore, Li and colleagues [15] identified binding motifs that are specific to each 14-3-3 isoform, and the RF/R/ARpSAPF was identified as the 14-3-3ε-specific binding motif. Babula and colleagues [16] reported PDB entries of 14-3-3–peptide complexes, showing that short peptides can bind to 14-3-3 proteins. The peptides occupy an amphipathic binding groove of 14-3-3 proteins [17,18,19] (Figure 1). Moreover, specific residues in the binding groove of 14-3-3 proteins that interact with the phosphorylated amino acid residue in the binding motif of 14-3-3 binding partners have been identified [18,19,20]. Specifically, Holmes and colleagues [21] identified Lys^50^, Arg^57^, Arg^130^, and Tyr^131^ residues of 14-3-3ε that interact with pSer/pThr of CDC25A in the 14-3-3ε–CDC25A interactions.

14-3-3 proteins have been implicated in various medical conditions, including metabolic disorders like obesity, where 14-3-3ζ promotes adipogenesis by regulating the activity of master adipogenic factors, peroxisome proliferator-activated receptor gamma (PPAR-γ), and CCAAT/enhancer binding protein (C/EBP) α [22]. Likewise, 14-3-3 proteins have also been associated with diabetes, where they regulate β cell function [23]. The 14-3-3 proteins are associated with various cancers, as discussed by Fan and colleagues [24]. In osteosarcoma, 14-3-3 proteins suppress transcription factor activity in the nucleus, thereby suppressing apoptosis [25]. Additionally, expression levels of 14-3-3 proteins are linked to the development of colon cancer [25]. Overexpression of the 14-3-3ε isoform is associated with abnormal growth of renal tumor cells and progression of renal cell carcinoma [26]. In cutaneous squamous cell carcinoma (cSCC), the 14-3-3ε isoform is overexpressed and mislocalized from the nucleus to the cytoplasm. In the cytoplasm of cSCC cells, the cell division cycle 25 A (CDC25A) protein expression is also increased, and its interaction with 14-3-3ε suppresses cancer cell apoptosis [21,27].

The 14-3-3 proteins can be pharmacological targets, and various compounds have been developed to target their interactions in signaling pathways involved in cancers and other diseases. Using computational methods, Corradi and colleagues [28] discovered BV01 and BV101 as inhibitors of 14-3-3σ – c-Abl interactions that have anti-proliferative activity in chronic myeloid leukemia. BV02 [29] and its derivatives [30] are among other inhibitors of 14-3-3 – c-Abl interaction that induce apoptosis in leukemia. Ballone and colleagues employed molecular dynamics (MD) simulations to discover small molecules targeting 14-3-3ζ–SOS interactions in cancer [31]. Additionally, Milroy [32] and Andrei [33] developed inhibitors of 14-3-3τ – protein interaction with therapeutic potential for Alzheimer’s disease.

In prior work in our laboratory, we developed two phosphopeptides derived from the two binding regions of CDC25A, residues 502–515 (pT) and residues 173–186 (pS), to 14-3-3ε, and both peptides induce cell death of sCCC cells [21]. We further optimized the pT peptide to improve its binding affinity for 14-3-3ε [34]. Here, we investigated the optimal pS sequence that binds 14-3-3ε. First, we shortened the pS peptide from 14 to 9 residues (pS(174–182)); then, based on the 14-3-3 binding motif [11], we modified the pS(174–182) at various positions. We used in silico and biophysical methods to study the binding of the pS analogs to 14-3-3ε.

## 2. Results

### 2.1. Computational Design of the Peptide Analogs

Inhibiting 14-3-3ε–CDC25A interactions is a promising way to promote apoptosis in cSCC. Our previously designed pS peptide inhibits 14-3-3ε–CDC25A interaction and induces death of cSCC cells with an IC_50_ of 29 μM [21]. In this work, we optimized the structure of the pS peptide to improve its binding affinity for 14-3-3ε. Initially, we sequentially truncated the pS peptides from both *N*- and *C*-termini to generate its analogs of various lengths (Table 1). Then, the stability of each peptide analog in complex with 14-3-3ε was studied in a 500 ns MD simulation. RMSD of Cα-atoms showed that all 14-3-3ε–peptide complexes experienced structural changes during the simulations, with RMSD of 0.07–0.25 nm, but no dissociation was observed for any peptide analog (Figure 2A and Appendix A). The interaction energies between the sequentially truncated peptides and 14-3-3ε changed from −1464.72 kJ/mol to −1165.30 kJ/mol (Appendix A). Given that the shortest analog, nine amino acid residues (pS(174–182)), bound to 14-3-3ε throughout the simulation, and its interaction energy was still largely negative, this peptide structure was used for further modifications.

Next, using the MM/PBSA method [35], computational Ala scannings were performed to ascertain residues of the pS(174–182) that contribute to the peptide binding to 14-3-3ε. All the Ala-substituted residues contributed to the peptide binding as indicated by the positive change in free energy (ΔΔG) (Figure 3 and Appendix A). Ala substitution of Gln^174^, Gln^176^, Asn^177^, and Arg^182^ resulted in ΔΔG in the range of 10.59–26.33 kJ mol^−1^. Moreover, Ala substitution of Arg^175^ showed a higher change in ΔΔG of 118.75 kJ mol^−1^, and pSer^178^ was shown to be the most important residue since its substitution with Ala resulted in ΔΔG of 1018.5 kJ mol^−1^.

Analysis of interactions between 14-3-3ε and pS(174–182) indicated that Arg^175^ and Arg^182^ residues of the peptide formed ionic interactions with specific residues of the protein (Appendix A). In addition, pSer^178^, Asn^176^, Ala^181^, Asn^177^, and Arg^182^ residues of the peptide formed H-bonds with various residues of 14-3-3ε. pSer^178^ formed the most persistent H-bonds with Arg^130^, Tyr^131^, Arg^50^, and Ala^179^ residues of 14-3-3ε (Appendix A). Based on the binding motifs of 14-3-3 proteins (Table 2), both P-1 and P+1 sides of pSer can be any amino acid.

Additionally, binding motif 2 suggests a preference for either Tyr or Phe in position P−2 (Table 2). Thus, we substituted amino acid residues in the sequence of pS(174–182) peptide with either aromatic or positively charged amino acid residues in P−2, P−1, and P + 1 positions to obtain various analogs of pS(174–182) (Table 3). The ability of these peptides to bind 14-3-3ε was assessed using MD simulations and biophysical methods.

### 2.2. MD Simulations of 14-3-3ε–Peptide Complexes

The stability of 14-3-3ε–peptide complexes and peptide conformation were assessed in MD simulations. RMSD of Cα-atoms indicated that in the 14-3-3ε–peptide complexes, the 14-3-3ε and the peptide in each complex experienced structural rearrangements, and all peptides bound the protein throughout the simulations (Figure 2A,B and Appendix A). The Cα-atom RMSD of complexes and protein in all complexes fluctuated between 0.08 nm and 0.25 nm, while the Cα-atom RMSD of the peptide fluctuated between 0.03 nm and 0.27 nm. [Tyr^176^]pS(174–182) and [Phe^179^]pS(174–182) peptides had the most structural rearrangement in the complexes (Appendix A). In all cases, peptides had large fluctuations around ~420 ns, but they still remained in the complex. Configurational entropy analysis (Figure 2D) showed that the 14-3-3ε–pS(174–182) system went through a sudden decrease in entropy between 0 ns and ~45 ns, a slow decrease between 45 ns and 300 ns, and then the system maintained equilibrium thereafter. Analysis of convergence indicated that the length of the simulation was sufficient to achieve conformational convergence [36,37], as indicated by an overlap of the successive sampled subspace at every 2 ns with the configuration space of the entire simulation (Figure 2D). Define Secondary Structure of Proteins (DSSP) analysis of the 14-3-3ε–pS(174–182) complex trajectory showed that 14-3-3ε had minimal structural changes during the simulation, while the peptide structure was predominantly random coil with some β-bend conformations throughout the simulation (Figure 2F). All the residues had coil structure except Gln^176^, which was in a β-bend conformation in the first 100 ns, after which it transitioned to a coil structure, and Ala^179^, which alternated between random coil and β-bend conformations throughout the simulation. The structural stability of the pS(174–182) peptide was further assessed in a 1000 ns peptide-only simulation. The Cα-atom RMSD of the peptide oscillated between 0.1 nm and 0.45 nm (Figure 2C). Unlike the peptide in the 14-3-3ε–pS(174–182) complex, the peptide alone had significant changes in structures between coils, β-bend, and β-turn conformations, as shown by DSSP analysis (Figure 2G). This indicates that the structure of the peptide is stabilized upon binding to 14-3-3ε. Although the peptide alone experienced structural changes during the simulation, overall, the random coil was the most predominant structure (Appendix A).

### 2.3. Binding Free Energy of the 14-3-3ε–Peptide Complexes

To determine binding free energies (ΔG_b_) of the peptides, we carried out 4.5 ns steered MD (SMD) simulations of 14-3-3ε–peptide complexes. Snapshots from SMD simulation trajectories were used as starting configurations for umbrella sampling (US) simulations. The trajectories of US simulations were analyzed by weighed histogram analysis (WHAM) [38] to determine the potential of mean force (PMF) from which ΔG_b_ is derived. During the SMD simulation of the 14-3-3ε–pS(174–182), a rupture event was observed around 900 ps, with a rupture force of 1000 kJ mol^−1^ nm^−1^, as indicated by the force–time curve (Figure 4A). All other peptides had double rupture events with rupture force between 600 kJ mol^−1^ nm^−1^ and 900 kJ mol^−1^ nm^−1^, except the Phe^177^-substituted peptide, which had a rupture event with a force of 800 kJ mol^−1^ nm^−1^ (Appendix A). The distance–time curve revealed no significant change in the distance between the center of mass (COM) of the protein and COM of the peptide in the first ~ 700 pS, with a progressive increase in the distance for the remaining simulation time (Figure 4B and Appendix A). All the umbrella histograms from the US simulations are overlapping (Figure 4C and Appendix A), indicating a sufficient number of samples. The PMF of the peptide movement along the ζ reaction coordinate (Figure 4D) steeply increased until the COM of 1.3 nm, after which it became constant, indicating peptide dissociation. All peptides bound 14-3-3ε with the negative free energy of binding ΔG_b_ (Figure 4E, Appendix A). pS(174–182) had ΔG_b_ of −128.62 ± 8.05 kJ mol^−1^, while the highest ΔG_b_ was observed from [Tyr^176^]pS(174–182), ΔG_b_ of −157.23 ± 6.57 kJ mol^−1^. Other analogs had ΔG_b_ between −130.39 ± 5.43 and −148.756 ± 5.855 kJ mol^−1^.

### 2.4. Secondary Structure of the Peptide Analogs

To ascertain secondary structures of pS(174–182) and its analogs, electronic circular dichroism (ECD) spectropolarimetry of the peptides was carried out with peptides dissolved in water or in 30%, 50%, or 75% 2,2,2-trifluoroethanol (TFE) in water.

Analysis of ECD spectra of the peptides in water showed a minimum of around 198 nm (Figure 5 and Appendix A), indicating that the peptides are in random coil conformation [39,40]. In aqueous TFE, the structures of the peptides did not show significant changes, except for pS, although there is an increase in the ~198 signal with increasing TFE concentration. The 14 amino acid residue peptide pS showed a red-shift in the ~198 minimum, and there was an emerging minimum at ~220 nm with an increasing percentage of TFE (Figure 5A), indicating some propensity to form an α-helical structure [39].

### 2.5. Experimental Validation of Binding of the Peptides to 14-3-3ε

The binding of the peptides to 14-3-3ε was further assessed using differential scanning fluorometry (DSF) and surface plasmon resonance (SPR) measurements. DSF was used to determine the effects of the peptides on the melting temperature (T_m_) of 14-3-3ε (Table 4, Figure 6 and Appendix A). The pS peptide caused an increase in melting temperature (ΔT_m_) of 0.35 ± 0.16 °C, while pS(174–182) induced no ΔT_m_. The Phe and Tyr substitutions in position 176 led to ΔT_m_ of 1.08 ± 0.34 °C and 1.10 ± 0.32 °C, respectively. Lys, Phe, and Tyr substitutions in position 179 led to ΔT_m_ of between 0.00 and 0.23 °C (Table 4). To quantitatively determine the affinity of the peptides for 14-3-3ε protein using SPR, His-tagged protein was immobilized on a nitrilotriacetic acid (Ni-NTA) sensor chip, and then binding of the peptides analogs to the protein was determined.

Sensograms of the peptides binding to 14-3-3ε and their respective binding isotherms are shown in Figure 7 and Appendix A. All the peptides bound to 14-3-3ε, although with different affinities (Table 4). Steady-state model fitting showed that the pS peptide bound 14-3-3ε with a K_D_ value of 0.22 ± 0.01 μM, while the pS(174–182) had a K_D_ of 6.56 ± 2.16 μM. Introducing Phe and Tyr in position 176 led to a K_D_ of 0.36 ± 0.63 μM and 0.41 ± 0.10 μM, respectively. Lys^177^, Phe^176^, and Tyr^176^ substitutions resulted in K_D_ of 5.23 ± 0.47 μM, 58.28 ± 4.67 μM, and 115.53 ± 16.7 μM, respectively. All peptides bound favorably to 14-3-3ε as demonstrated by the negative free energy of binding (ΔG_b_) derived from K_D_ (Table 4).

## 3. Discussion

Inhibiting 14-3-3ε–CDC25A interaction in cSCC using CDC25A-derived phosphopeptides is an approach that is supported by our previous work [21] documenting interference of pS peptide with the 14-3-3ε – CDC25A interaction and increased cell death of cSCC cells with a relatively high IC_50_ (29 µM). Here, we report our efforts to optimize the structure of the pS peptide to improve its binding to 14-3-3ε. We designed peptides in silico, and we used computational and biophysical methods to study their binding to 14-3-3ε. We used two approaches to obtain the pS analogs. First, we shortened the peptide from 14 to 9 amino acid residues; then, based on the proposed binding motifs of 14-3-3 proteins (Table 1), we modified the peptide in positions 176, 177, and 179 to obtain its analogs. The binding motifs of 14-3-3 proteins are defined by 6 to 7 amino acid residues (Table 1). We previously showed that a shorter phospho-threonyl peptide bound 14-3-3ε with higher affinity than its longer analog [34]. Shortening a peptide generally improves its enzymatic stability since fewer peptide bonds are available for degradation. Furthermore, the eventual removal of the Met residue from the parent peptide (Table 1) improves the oxidative stability of the resultant analog. Thus, we set out to find a shorter analog of the pS peptide. RMSD analysis of the sequentially truncated peptide indicated that the shortest nine-amino-acid residue peptide (pS(174–182)) formed a stable complex with 14-3-3ε. Furthermore, DSSP analysis showed that the pS(174–182) bound 14-3-3ε, as indicated by a more stable secondary structure of the peptide when in complex with 14-3-3ε compared to the peptide alone (Figure 2F). Therefore, we further explored the role of pS(174–182) peptide residues in 14-3-3ε binding. Computational Ala scanning of residues of the peptide indicated that pSer^178^ is the most important residue for pS(174–182) binding to 14-3-3ε, which is in agreement with H-bond analysis showing that pSer has the longest-lasting interactions (Appendix A). Arg^175^ is another residue that highly contributes to the binding of 14-3-3ε. This is due to ionic interactions between Arg and Glu^134^, Glu^183^, and Asp^226^ of 14-3-3ε (Appendix A). The many interactions of Arg^175^ with 14-3-3ε confirm the proposition by the binding motif 1 (Table 2) that Arg is preferred in the P-3 position.

The primary structure of the pS peptide fits in the proposed binding motifs of 14-3-3 proteins (Table 1). On the basis of the two binding motifs for 14-3-3 proteins, we introduced aromatic amino acid residues at positions 176, 177, and 179 of pS(174–178) to target various interactions involving aromatic amino acid residues. Aromatic rings are an important part of protein–drug and protein–protein interactions [41] due to their ability to form various non-covalent interactions, including π–π [41], anion–π [42], cation–π [43], and amide–π [44] interactions.

Using SMD and US simulations, we determined the ΔG_b_ of the designed peptide analogs. The large negative ΔG_b_ for pS(174–182) indicates favorable binding to 14-3-3ε. Introducing aromatic amino acid residues in positions 176 and 179 resulted in peptides with more favorable binding than the parent pS(174–182). This improvement in peptide binding may be due to the various interactions formed between aromatic residues and amino acid residues of 14-3-3ε. In position 177, Tyr and Phe substitutions resulted in ΔG_b_ that is comparable to that of pS(174–182). These results indicate that aromatic amino acid residues are not more favorable for binding in this position over the native residue; hence, the [Tyr^177^]pS(174–182) and [Phe^177^]pS(174–182) peptide analogs were disqualified from further studies. We also introduced Lys amino acid residue in the 177 position to target interactions involving positively charged amino acid residues, which resulted in a peptide with more favorable binding than that of pS(174–182).

Secondary structures of the peptide determined by ECD spectropolarimetry indicated that shortening the peptide and modifying the short peptide did not affect its secondary structure, as shown by all the peptides having random coil structures in water. These results are in agreement with the DSSP analysis of pS(174–182), which revealed that the peptide is a predominantly random coil (Figure 2G and Appendix A). Also, shortening the peptide reduced its tendency to form α-helical structures at high TFE concentrations. TFE mimics a dielectric environment inside a protein; thus, the induced α-helical conformation is not favored for binding. The higher affinity of the pS peptide, compared to its 9-mer analogs, however, is most likely due to interactions of pS with 14-3-3ε through hydrophobic interactions. In our previous study [34], similar results were obtained. The 14-mer, pT peptide exhibited α-helical conformation in TFE; however, this peptide has lower affinity than those of its 9-mer analogs that exhibit random coil conformation.

DSF results indicated that the pS peptide caused ΔT_m_ of 0.35 ± 0.16 °C. At the studied concentration, pS(174–182) did not cause any change in melting temperature. The Phe and Tyr modifications in the 176 position led to a larger ΔT_m_ (Table 4), while other modifications resulted in minimal ΔT_m_. For DSF, we used a 30 μM peptide concentration, the concentration at which our pT analogs showed increased T_m_ for the complexes [34]. The parent pS peptide also induced a shift in T_m_ at 30 µM. Thus, all the peptide analogs were evaluated at this concentration. SPR studies indicated that the pS peptide bound 14-3-3ε with a K_D_ of 0.22 ± 0.01 μM, while the pS(174–182) had a K_D_ of 6.56 ± 2.16 μM, in agreement with the DSF results that shortening the peptide reduced its binding to 14-3-3ε. This indicates that some of the truncated amino acid residues of the pS are important for binding the peptides to 14-3-3ε. These residues most likely contribute to binding through hydrophobic effects. Substitution of Ala^179^ by Tyr or Phe did not improve the binding of the peptide, indicating that replacing the side chain -CH_3_ group of Ala with a bulkier aromatic ring introduces steric constraints to binding. Phe^176^ and Tyr^176^ modifications in pS(174–182) rescued binding affinity of the peptide (K_D_ 0.36 ± 0.63 μM and 0.41 ± 0.10 μM, respectively). This improvement in affinity could be due to the aromatic amino acid residues forming various interactions like π–π, amide–π, cation/anion–π, and CH–π with residues of 14-3-3ε (Appendix A). All other modifications did not improve the affinity of pS(174–182). The Phe^176^ and Tyr^176^ modifications results are in agreement with binding motif 2 and show that either Phe or Tyr are favored amino acid residues in the P−2 position to interact with 14-3-3ε. The differences between the PMF-derived (Appendix A) and the experimentally determined binding free energies (Table 4) can be attributed to the fact that PMFs were derived from one-dimensional SMD simulations. Nevertheless, among the shortened peptides, [Phe^176^]pS(174–182) experimentally has the second best K_D_.

CDC25A binds 14-3-3 proteins through two separate regions that contain either pSer or pThr amino acid residues [27]. We used 14-mer phosphopeptides corresponding to these two regions and revealed that these peptides bind 14-3-3ε, interrupt CDC25A–14-3-3ε interaction, and thus induce death of cSCC cells [21]. To the best of our knowledge, our group was the first to show that these peptides interfere with CDC25A activity. In an effort to improve binding affinities of the peptides for 14-3-3ε, we have demonstrated that the pThr-containing 9-mer peptide has a higher affinity for 14-3-3ε than its 14-residue parent peptide [34], while among pSer-containing analogs, the 14-mer had the highest affinity. Overall, the pThr-containing peptides are better binders of 14-3-3ε than the pSer-containing peptides. Although the sequence of the pT(502–510) peptide does not fit in the general binding motif pattern of 14-3-3 proteins, it still exhibited a higher affinity for 14-3-3ε than the pS(174–182) that fits in the general binding motifs pattern. Thus, the pT(502–510) peptide will likely be more selective for the 14-3-3ε isoform. We have now developed two classes of phosphopeptides that bind with high affinity to 14-3-3ε. However, their intracellular availability, which will be determined in future studies, will be the deciding factor for their usage against cutaneous squamous cell carcinoma.

## 4. Materials and Methods

The methods presented herein are described in detail in our previous publication [34].

### 4.1. MD Simulations

#### 4.1.1. 14-3-3ε–Peptide Complexes Preparation

The initial structure of the 14-3-3ε–peptide complex, used in in silico studies, was obtained from the previously published results of 14-3-3ε interactions with CDC25A-derived pS phosphopeptide [21]. Using the YASARA program [45], the peptide was truncated from both *N*- and *C*-terminal ends, one amino acid at a time, to generate peptide analogs of different lengths, the shortest being the nine-amino-acid residue peptide, pS(174–182). To preserve the electronic structure of the backbone as in the parent pS, all the designed peptides were *N*- and *C*-terminally acetylated and amidated, respectively. For the modified peptides, the initial peptide structure was obtained from MD simulations (described below) of the 14-3-3ε–pS(174–182) complex. For each peptide analog, modifications were made in the sequence of the pS(174–182) by substituting the amino acid residue of the peptide with desired amino acid residues, one modification at a time, using the YASARA program, version 24.1.23.

#### 4.1.2. MD Simulations of 14-3-3ε–Peptide Complex

The 14-3-3–peptide structures were obtained as described above, and MD simulations of the complex structures were carried out using the GROMACS-2021 [46] software package and CHARMM36m [47] force field. Each 14-3-3–peptide complex was solvated in a truncated dodecahedron box with a TIP3P water model [48]. To relax the system, two consecutive energy minimizations were performed. Then, the system was equilibrated in a 10 ns MD simulation at a constant number of molecules, pressure, and temperature (NPT) at 1 bar pressure and 310 K temperature using the Berendsen methods [49]. Thereafter, a 500 ns NPT production run simulation was performed for each system. For the peptide-only simulation, the starting structure was obtained by deleting 14-3-3ε from the 14-3-3ε–peptide complex using the YASARA program. The simulation was conducted as described above, except that the production run was for 1000 ns.

#### 4.1.3. Trajectory Analysis

The trajectory analysis was performed as previously described [34]. To assess if systems achieved equilibrium, configurational entropy [50] and convergence of the system were calculated using the *covar* module of GROMACS. To determine the structural stability of 14-3-3ε and the peptides throughout the simulations, their Cα RMSD was calculated using the *rms* module of GROMACS. Changes in the secondary structure of the peptide and protein were studied using the defined secondary structure of proteins (DSSP) method [51]. H-bonds and ionic interactions were calculated using H-bonds and salt bridges plugins, respectively, of VMD software, version 1.9.3 [52]. For H-bonds, the donor–acceptor distance was set to 0.4 nm, and the oxygen–nitrogen cut-off distance was set to 0.8 nm for salt bridges.

Configurational entropies were determined as before [34]. The trajectories were sampled at every 2 ns, and eigenvectors corresponding to the 150 highest eigenvalues were used for the calculations. Convergences of simulations were determined by following the method of Hess [36]. The covariance matrices of Cα-atom movement at every 2 ns were compared to that of the whole intervals of the simulation.

#### 4.1.4. Computational Ala Scanning and Free Energy Calculation Using the MM/PBSA Method

Using the last structure from the MD simulation of the 14-3-3ε–pS(174–182) complex (described above) as our input structure, 2 ns production simulation was performed. Similar parameters were used as described in MD simulations. Ala scanning of residues of the peptide was performed using the g_mmpbsa program [35], and free energy (ΔG) was calculated using the MM/PBSA method [35].
ΔG = ΔE_MM_ + ΔG_sol_ − TΔS(1)
where ΔG is the free energy of binding, ΔE_MM_ is the molecular mechanical binding energy, ΔG_sol_ is the free energy of solvation, and −TΔS is the interaction entropy contribution [53].
E_MM_ = E_bonded_ + E_nonbonded_(2)
where E_bonded_ is the bonded interactions, which include bond, angle, dihedrals, and improper interactions. The E_nonbonded_ is the nonbonded interactions; it includes both electrostatic and van der Waals interactions.
∆G_sol_ = ∆G_polar_ + ∆G_nonpolar_(3)

G_polar_ is the electrostatic contribution, and G_nonpolar_ is the non-electrostatic contributions to the solvation-free energy.
(4)−TΔS=RTln
ΔE^int^_pl_ = E^int^_pl_ − <E^int^_pl_>(5)
is the fluctuation of protein–ligand interaction energy around the ensemble averaged interaction energy, R is gas constant, T is the temperature in Kelvin, and β is the standard value of 0.92 kcal/mol.

For each Ala scan, ΔΔG was calculated as the difference between the ΔG when a residue is substituted with Ala and ΔG without any substitution.

#### 4.1.5. SMD and US Simulations

The input structure of 14-3-3ε–peptide complexes used in SMD simulations was obtained as described above (in the 14-3-3ε–peptide complex preparation section). The 4.5 ns simulations were performed using the GROMACS-2021 software package and CHARMM36m force field. The simulations were carried out as described before [34], except that the COM of the peptide was defined as COM of the pSer residue of the peptide, and COM of 14-3-3ε was defined as COM of Lys^50^, Arg^57^, Arg^130^, and Tyr^131^. For US simulations, 26–32 snapshots from SMD simulations were used as starting configurations.

### 4.2. Protein and Peptides

(His)_6_-14-3-3ε was from Novus Biologicals (Centennial, CO, USA). The peptides were synthesized by Biosynth International Inc. (Louisville, KY, USA). All peptides were purified to higher than 95% purity.

### 4.3. CD Spectrometry Measurements

Lyophilized peptide powders were dissolved in either nanopure water or 30%, 50%, and 75% (*v*/*v*) 2,2,2-trifluoroethanol (TFE) in water to a concentration of 100 μM, and spectra of each peptide were collected using a Jasco J-810 spectropolarimeter (Jasco Inc., Easton, MD, USA). All spectra were background-corrected, and each spectrum is an average of 20 scans. The spectra are presented as CD (mdeg).

### 4.4. Differential Scanning Fluorimetry (DSF)

Thermal unfolding of 14-3-3ε (2 µM) and its complex with peptides (30 μM) was monitored in the presence of a hydrophobic protein-binding dye, SYPRO-orange (Invitrogen, Carlsbad, CA, USA), using the BioRad CFX384 Touchreal-time PCR (Hercules, CA, USA). The assays were performed in 20 mM TRIS, 150 mM NaCl, 1 mM DTT, and pH 7.4 buffer. The first derivative of the melting curves was plotted using GraphPad Prism software, version 10.0.3 to determine T_m_ of the protein. ΔT_m_ was determined as the difference between T_m_ of 14-3-3ε in the presence of a peptide and T_m_ of 14-3-3ε alone.

### 4.5. Surface Plasmon Resonance (SPR)

The SPR analyses were carried out using the Biacore 8K surface plasmon resonance instrument (Cytiva, Marlborough, MA, USA). A total of 100 nM His_6_-14-3-3ε was immobilized on the NiCl_2_-activated NTA chip (Cytiva, Marlborough, MA, USA) as previously described [34]. Then, different concentrations of peptide in the range of 0−500 μM (depending on the peptide) were injected. The experiments were performed in 20 mM TRIS, 150 mM NaCl, 50 μM EDTA, 0.005% Tween 20, and pH 7.4 buffer. The affinities of the peptides binding to 14-3-3ε were determined by fitting the data to a steady-state binding model using Biacore Insight Evaluation Software (version 5.0.18.22102).

## 5. Conclusions

We designed shortened peptide analogs of the pS peptide that bind 14-3-3ε with high affinities. We used two in silico approaches to design the peptides. First, we shortened the peptide from 14 to 9 amino acid residues, and then, guided by the binding motifs of 14-3-3 proteins, we introduced aromatic amino acid residues to target various interactions involving aromatic rings. The binding of the designed peptide analogs was assessed using both in silico and biophysical methods. Our studies showed that shortening the pS peptide reduced its binding affinity to 14-3-3ε. However, the affinity was restored by substituting Gln^176^ with either Phe or Tyr amino acids. The pro-apoptotic activity of the peptides in cSCC cells will be studied in the future. The short peptide analogs, 9-amino-acid residues, [Phe^176^]pS(174–182), and [Tyr^176^]pS(174–182) peptides have a binding affinity that is comparable to that of the 14-amino-acid residues peptide, pS. However, the shorter peptides could have some advantages since they are most likely not as immunogenic as their longer peptide, and they are more likely to be more prototypically stable, given the fewer peptide bonds compared to the longer peptides.

## Figures and Tables

**Figure 1 ijms-25-04918-f001:**
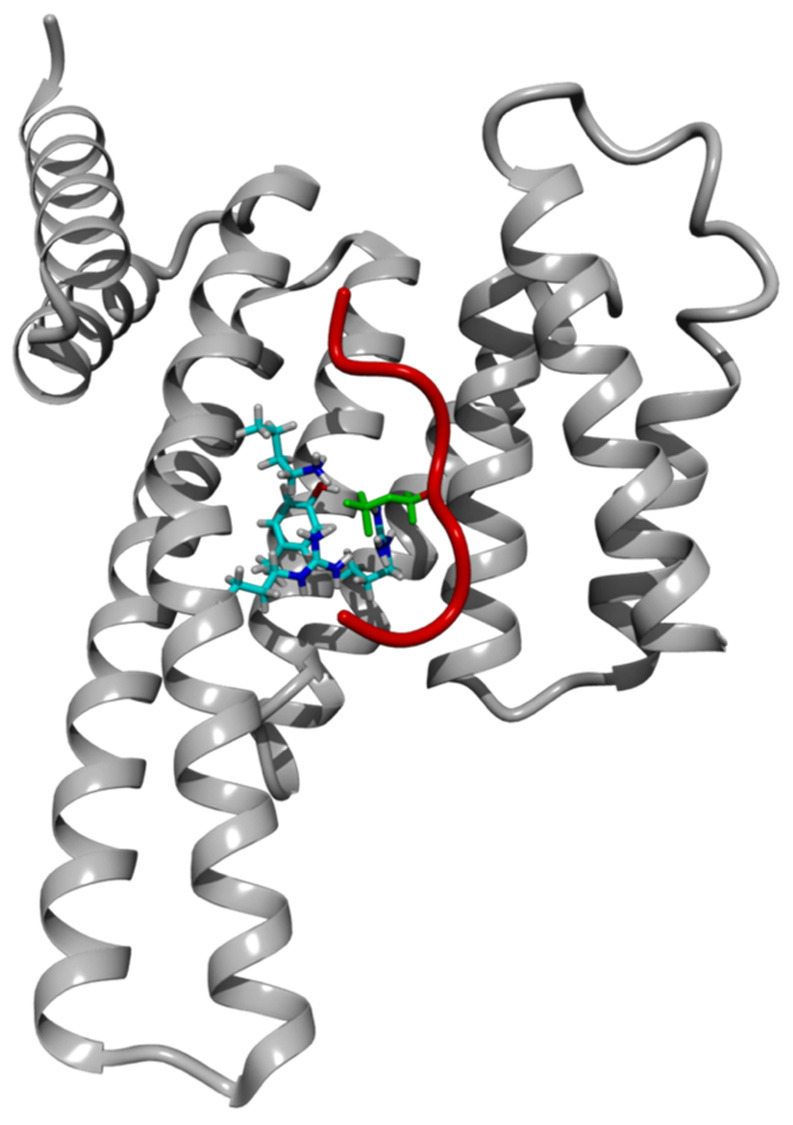
Structure of the 14-3-3ε–pS(174–182) complex. The structure is the largest cluster representative of the MD simulation of the complex. The monomer structure of 14-3-3ε is presented as a gray ribbon, and the peptide backbone is shown as a red tube. The pSer^178^ residue of the peptide is shown as a green stick, and Lys^50^, Arg^130^, and Tyr^131^ residues of 14-3-3ε that interact with pSer178 are shown as colored sticks.

**Figure 2 ijms-25-04918-f002:**
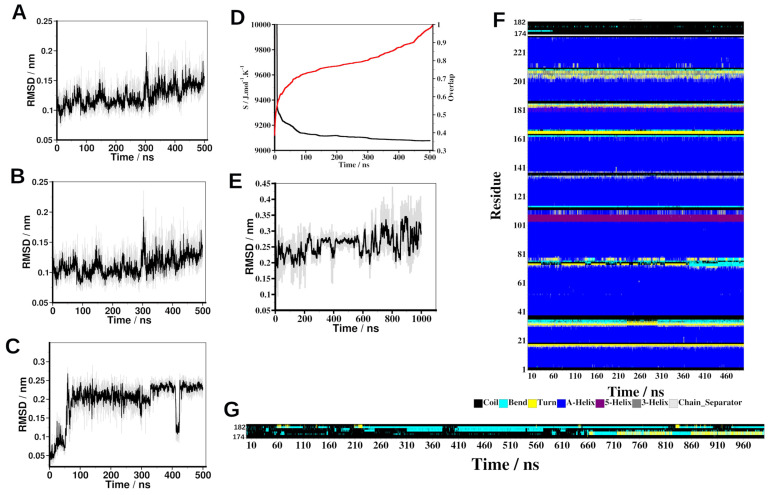
MD simulations of 14-3-3ε–pS(174–182) complex and pS(174–182) peptide alone. RMSD (**A**) 14-3-3ε – pS(174–182) complex; (**B**,**C**) 14-3-3ε and pS(174–182), respectively, from the protein–peptide complex simulation. (**D**) Configurational entropy (black) and configurational subspace overlap at every 2 ns of the sampled trajectory (red) of 14-3-3ε – pS(174–182) simulation system. (**E**) pS(174–182) peptide from a simulation of the peptide alone. Change in secondary structure of (**F**) 14-3-3ε – pS(174–182) complex, (**G**) pS(174–182) from a simulation of the peptide alone. The protein–peptide simulation was 500 ns, and the peptide-only simulation was 1000 ns.

**Figure 3 ijms-25-04918-f003:**
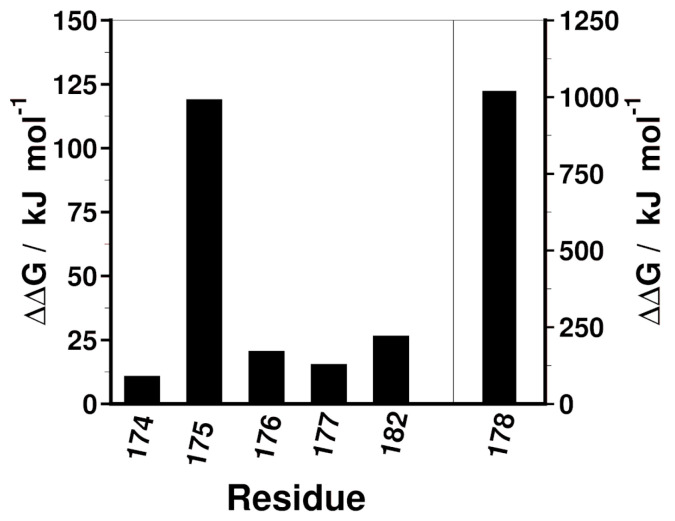
Contribution of side chain of amino acid residues of pS(174–186) to the peptide binding to 14-3-3ε. Change in free energy (ΔΔG) was calculated by Ala scan of the residues in a 2 ns simulation. The position 178 is plotted on a larger *y*-axis scale (right).

**Figure 4 ijms-25-04918-f004:**
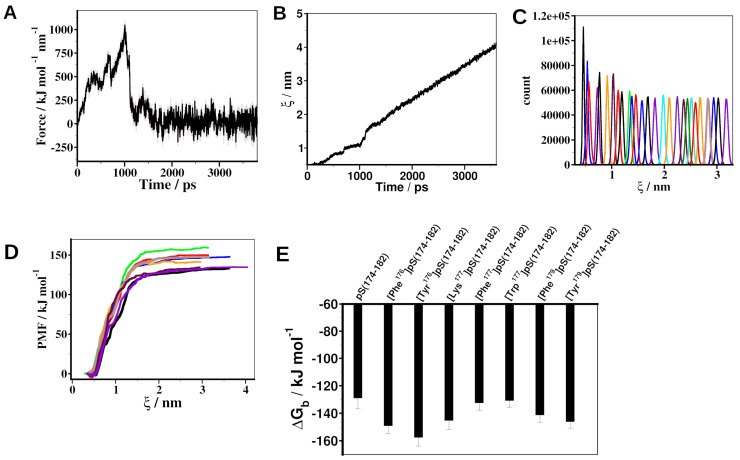
Steered MD and umbrella sampling simulations of 14-3-3ε–peptide complexes. (**A**) External force applied to pull the pS(174–182) peptide away from the binding pocket of 14-3-3ε during the simulation. (**B**) Distance between COM of protein and COM of peptide during the simulation. (**C**) Umbrella sampling histograms of configurations from a 30 ns simulation of 14-3-3ε − pS(174–182) complex simulations. (**D**) Potential of mean force (PMF) of 14-3-3ε–peptide complexes obtained by weighed histogram analysis. pS(174–182) (black); [Phe^176^]pS(174–182) (red); [Tyr^176^]pS(174–182) (green); [Lys^177^]pS(174–182) (blue); [Phe^179^]pS(174–182) (orange); [Tyr^179^]pS(174–182) (brown); [Phe^177^]pS(174–182) (cyan); [Trp^177^]pS(174–182) (purple). (**E**) Free energy of binding (ΔGb) of 14-3-3ε–peptide complex, derived from PMF.

**Figure 5 ijms-25-04918-f005:**
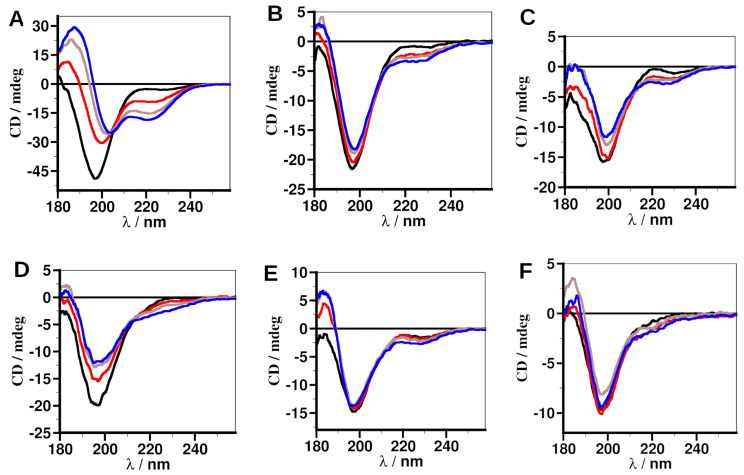
Electronic circular dichroism (ECD) spectra of pS peptide analogs. (**A**) pS, (**B**) pS(174–182), (**C**) [Phe^176^] pS(174–182), (**D**) [Tyr^176^] pS(174–182), (**E**) [Phe^179^] pS(174–182), (**F**) [Tyr^179^] pS(174–182). Black, water; red, 30% TFE; brown, 50% TFE; blue, 70% TFE.

**Figure 6 ijms-25-04918-f006:**
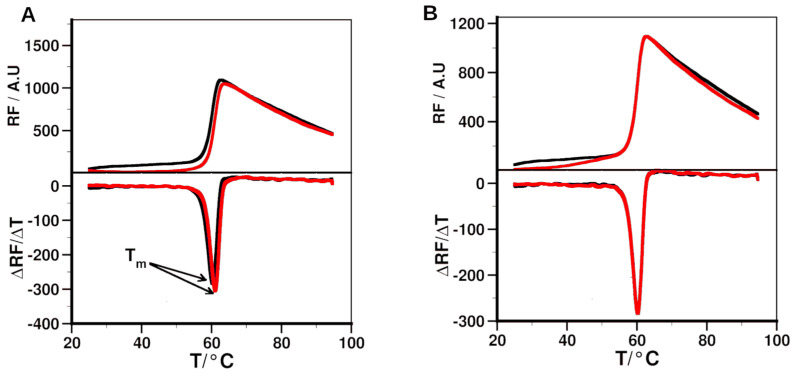
Thermal denaturation of 14-3-3ε and its complex with peptides determined by differential scanning fluorimetry. Melting curve of 14-3-3ε alone (black), and 14-3-3 in the presence of a peptide (red), and their respective first derivative curves. (**A**) pS, (**B**) pS(174–182). Effect of the peptide on the melting temperature (Tm) of 14-3-3ε was determined as the shift in melting temperature (ΔTm) of 14-3-3ε in the presence of a peptide.

**Figure 7 ijms-25-04918-f007:**
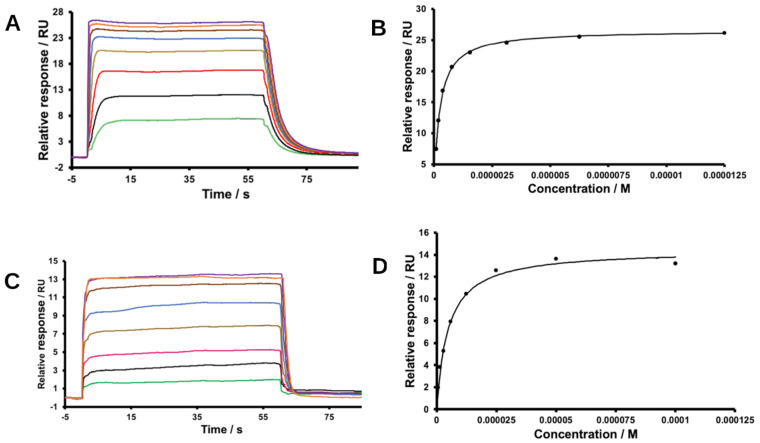
Representative surface plasmon resonance sensogram and calculated binding isotherm. 14-3-3ε was immobilized on an NTA sensor chip. Binding of the ligands to 14-3-3ε was measured at a concentration range of 0–100 μM. Sensograms and their respective binding isotherm of (**A**,**B**) pS and (**C**,**D**) pS(174–182).

**Table 1 ijms-25-04918-t001:** Amino acid sequence of sequentially truncated analogs of the pS peptide.

Peptide	Sequences *
pS	Ac-Thr-Gln-Arg-Gln-Asn-pSer-Ala-Pro-Ala-Arg-Met-Leu-Ser-Ser-NH_2_
pS(173−185)	Ac-Thr-Gln-Arg-Gln-Asn-pSer-Ala-Pro-Ala-Arg-Met-Leu-Ser-NH_2_
pS(173−184)	Ac-Thr-Gln-Arg-Gln-Asn-pSer-Ala-Pro-Ala-Arg-Met-Leu-NH_2_
pS(173−183)	Ac-Thr-Gln-Arg-Gln-Asn-pSer-Ala-Pro-Ala-Arg-Met-NH_2_
pS(174−183)	Ac-Gln-Arg-Gln-Asn-pSer-Ala-Pro-Ala-Arg-Met-NH_2_
pS(174–182)	Ac-Gln-Arg-Gln-Asn-pSer-Ala-Pro-Ala-Arg-NH_2_

* pSer, phosphoserine; Ac and -NH_2_, acetyl and amide, respectively, protecting groups.

**Table 2 ijms-25-04918-t002:** Binding motifs of 14-3-3 proteins and sequence of pS(174–182) peptide.

Motifs		P−4	P−3	P−2	P−1	P	P+1	P+2			
Motif1			Arg	Ser	Xaa	pSer/Thr	Xaa	Pro			
Motif 2		Arg	Xaa	Tyr/Phe	Xaa	pSer/Thr	Xaa	Pro			
pS(174–182)	Ac	Gly	Arg	Gln	Asn	pSer	Ala	Pro	Ala	Arg	NH_2_

**Table 3 ijms-25-04918-t003:** Peptide analogs of Ac-CDC25A(174–182)-NH2.

Peptide	Sequences *
pS	Ac-Thr-Gln-Arg-Gln-Asn-pSer-Ala-Pro-Ala-Arg-Met-Leu-Ser-Ser-NH_2_
pS(174–182)	Ac-Gln-Arg-Gln-Asn-pSer-Ala-Pro-Ala-Arg-NH_2_
[Phe^176^]pS(174–182)	Ac-Gln-Arg-Phe-Asn-pSer-Ala-Pro-Ala-Arg-NH_2_
[Tyr^176^]pS(174–182)	Ac-Gln-Arg-Tyr-Asn-pSer-Ala-Pro-Ala-Arg-NH_2_
[Lys^177^]pS(174–182)	Ac-Gln-Arg-Gln-Lys-pSer-Ala-Pro-Ala-Arg-NH_2_
[Phe^177^]pS(174–182)	Ac-Gln-Arg-Gln-Phe-pSer-Ala-Pro-Ala-Arg-NH_2_
[Trp^177^]pS(174–182)	Ac-Gln-Arg-Gln-Trp-pSer-Ala-Pro-Ala-Arg-NH_2_
[Phe^179^]pS(174–182)	Ac-Gln-Arg-Gln-Asn-pSer-Phe-Pro-Ala-Arg-NH_2_
[Tyr^179^]pS(174–182)	Ac-Gln-Arg-Gln-Asn-pSer-Tyr-Pro-Ala-Arg-NH_2_

* Ac and -NH_2_, acetyl and amide, respectively, protecting groups; substitutions are marked with red color.

**Table 4 ijms-25-04918-t004:** Melting temperature shift (ΔT_m_) and binding affinity of peptide analogs for 14-3-3ε. ΔT_m_ values are an average ± SD of *n* ≥ 5, and K_D_ values are average ± SD of *n* ≥ 3. The free energies of binding (ΔG) were derived from K_D_.

Peptide	ΔT_m_/°C	K_D_ ± SD/μM	ΔG ± SD/kJ mol^−1^
pS	0.35 ± 0.16	0.22 ± 0.01	−37.96 ± 0.09
pS(174–182)	0	9.05 ± 0.83	−28.76 ± 0.23
[Phe^176^]pS(174–182)	1.08 ± 0.34	0.36 ± 0.63	−36.48 ± 0.63
[Tyr^176^]pS(174–182)	1.10 ± 0.32	0.41 ± 0.10	−36.76 ± 0.46
[Lys^177^]pS(174–182)	0.23 ± 0.09	5.23 ± 0.47	−30.12 ± 0.23
[Phe^179^]pS(174–182)	0	58.28 ± 4.67	−24.15 ± 0.2
[Tyr^179^]pS(174–182)	0	115.53 ± 16.7	−22.47 ± 0.39

## Data Availability

All experimental data are available in the manuscript and in the Appendix A.

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
