# Peer review of "The Development of CDC25A-Derived Phosphoseryl Peptides That Bind 14-3-3ε with High Affinities"

_ijms, 2024, doi:10.3390/ijms25094918_

Round 1

Reviewer 1 Report

Comments and Suggestions for Authors

The authors present a report on the design of peptide analogs of pS that exhibit high affinities for binding to 14-3-3ε. The peptide derived from CDC25A, PS, inhibits the interaction between 14-3-3ε and CDC25A, offering a promising approach to promote apoptosis in cSCC. In this study, the peptide was shortened to a 9-residue pS(174-182) peptide, and modification were introduced at positions 176, 177 and 179. The interactions between those pS analogs and 14-3-3ε were subsequently analyzed using a combination of computational and biophysical methods, including steered molecular dynamic, free energy calculations, electronic circular dichroism, differential scanning fluorimetry, etc. The results suggest that pSer178 is crucial for the binding of pS(174-182) to 14-3-3ε, and the introduction of aromatic amino acid at residues 176 and 179 can further improve the binding. Those analogs maintained a random coil structure in water, similar to pS(174-182). Notably, the [Phe176]pS(174-182) and [Tyr176]pS(174-182) peptides exhibit comparable binding affinity to pS, providing a structural foundation for future drug design against cSCC.

Overall, the manuscript is well-structured and presents interesting results. I believe the manuscript is suitable for publication after minor revisions.

1.     Line 39, “to14-3-3”, a space is missing.

2.     Figure 5, line 205, “blue, 30 % TFE” should be “red, 30 % TFE”.

3.     The authors could consider swapping the second and third paragraphs of the introduction.

Author Response

  1. Line 39: Inserted space between "to" and "14-3-3"
  2. Figure 5, corrected the mistake
  3. Swapping the second and 3rd paragraph: The 1st and second paragraph give a functional and structural introduction to 14-3-3 proteins which followed, in the 3rd and 4th paragraph, the abnormal physiology and pharmacology of the proteins.
    Therefore, swapping the paragraphs would break this flow of our introduction.   

Reviewer 2 Report

Comments and Suggestions for Authors The research titled "The Development of CDC25A-Derived Phosphoseryl Peptides That Bind 14-3-3ε With High Affinities" by Seraphine Kamayirese and co-workers is Very nicely done work presented in very nice manner. i have only one suggestion, please add some results from literature and compare with similar work reported by others. 

Author Response

In the introduction, from Line 68, we refer to work of other groups to describe peptide (refs 28 and 29) and non-peptidic  inhibitors(refs 30-33). To our best knowledge, no other group developed, using rational design, phosphopeptide inhibitors of 14-3-3epsilon interactions. Especially, no other groups targeted CDC25A - 14-3-3epsilon interactions, therefore, we cannot add similar results from literature and compare those to ours.

Reviewer 3 Report

Comments and Suggestions for Authors

The manuscript of Seraphine Kamayirese and coauthors is devoted studying the short peptides inhibiting the interaction between 14-3-3ε protein and CDC25A phosphatase using computational and biophysical methods.

My remarks relate mainly to the presentation of methods and results.

1.      The methods used should be understandable to readers without references to previous works.  What was the initial structure of peptides and complexes? What was the charge of the peptides, pH, and ionic strength of the solution?

2.       It is necessary to explain in detail how the configurational entropy was calculated. What parameters and criteria were used?

3.      RMSD analysis is a standard but not very informative method used to confirm the stability of the complex. I recommend to replace RMSD data in Figure 1 with interaction energies between protein and peptides. In this case, readers can evaluate which peptide interacts more strongly than others additionally to stability of the complexes.

4.      The calculated PMF values do not correlate with the experimental data. It will be interesting to see a comparison of the protein-peptide interaction energies obtained experimentally and theoretically and a discussion of possible reasons for non-coincidence.

5.      MM/PBSA method was used for Ala-scanning of residues of the pS(174-182). I recommend to show in the table the contribution of components and especially the entropy contribution to the free energy of binding.

Minor remarks:

A small number of typos were found. For example, in Fig. 5 it is written “water; blue, 30 % TFE; brown, 50% TFE; blue, 70 % TFE.”

Figure 1 is very dark.

References should be checked.

Comments on the Quality of English Language

The article is written in good English.

Author Response

  1. - As an accepted standard in the scientific literature, already published and detailed methods are not repeatedly reported over and over again. Following this standard, we refer to our detailed methods described in our previous publication (ref 34).   
    - The net charges of peptides were not guiding factor in our design because, it is not trivial to predict a residue charges of peptides inside a binding pocket in a dehydrated and low dielectric environment.  
    - The initial primary structures of peptides are given in Table 1 and the cartoon representation 14-3-3epsilon - pS(174-182) complex structure is given in figure 1. 
    - "pH and ionic strength of the solution?"  
    for the DSF buffer, at line 422 of the original manuscript we inserted: 
    "The assays were performed in 20 mM Tris, 150 mM NaCl, 1 mM DTT, pH 7.4 buffer."
    for the SPR buffer, at line 431 of the original manuscript we inserted the following:
    "The experiments were performed in 20 mM Tris, 150 mM NaCl, 50 μM EDTA, 0.005% Tween 20, pH 7.4  buffer."

  2. To expand on the configuration entropy calculation, the following were inserted at line 367 of the original manuscript: 
    "The trajectories were sampled at every 2 ns and eigenvectors corresponding to the 150 highest eigenvalues were used for the calculations."

  3. We followed the reviewer' suggestion and calculated the interaction energies between the sequentially truncated peptide and 14-3-3epsilon and included the data as  supplementary table S1. 
    Also, at line 93 of the original manuscript inserted the following:
    " The interaction energies between the sequentially truncated peptides and 14-3-3ε changed from -1464.72 kJ/mol to -1165.30 kJ/mol (Table S1.)"
    and in the following sentence after  "...simulation":
    ", and its interaction energy was still large negative, " 

    The subsequent supplementary tables were renumbered accordingly.

  4. "The calculated PMF..." Indeed the free energy of binding from PMF and experimentally data are not agreeing. This is attributed to the fact that the SMD (pulling) simulations are  one-dimensional (zeta reaction coordinate). Furthermore during the subsequent US simulations  the peptide positions were restrained. Therefore this methodology never give perfect match for experiental data.  Nevertheless, we used PMF calculations to guide our design of synthetic peptides.   The experimentally best binder Phe176 analog peptide is second best binder from the PMF calculations.

  5. Following the reviewer's suggestion, we created a supplementary table 2 listing the contributing components to the MM/PBSA analysis.
    In line 102 of the original manuscript after "Figure 3" we inserted " and Table S2".

    - Minor remarks: we corrected the typos as indicated already in the response for reviewer 1.
    - We replaced Fig 1. with a revised, much lighter figure. 
    - checked the references and and the wrongly indicated ref 38 was modified to 36.

Reviewer 4 Report

Comments and Suggestions for Authors

File attached

Author Response

1.  ” ...must emphasize more clearly what advantages the short peptide analogs..” In the first paragraph of the discussion, at line 253 of the original manuscript we inserted the followings:

“Shortening a peptide generally improves its enzymatic stability since less number of peptide bonds are available for degradation. Furthermore, the eventual removal of the Met residue from the parent peptide (Table 1) improves oxidative stability of the resultant analog.“

   - "pT(502-510) was already proposed bye the authors...":

At the end of the last paragraph of the discussion we inserted the the following:

"We have now developed two different classes of phosphopeptides that bind with high affinity to 14-3-3ε. However, their intracellular availability, which will be determined in future studies, will be the deciding factor for their usage against cutaneous squamous cell carcinoma."

2.  The difference between calculated and experimental binding free energies can be attributed to the fact that the SMD (pulling) simulations are  one-dimensional (zeta reaction coordinate). Furthermore during the subsequent US simulations  the peptide positions were restrained. Therefore, this methodology never give perfect match for experimental data.  Nevertheless, we used PMF calculations to guide our design of synthetic peptides.  The experimentally best binder Phe176 analog peptide is second best binder from the PMF calculations.

In the discussion at the end of the last before paragraph, line 323 of the original paragraph, we inserted the following:

"The differences between the PMF-derived (Table S4) and the experimentally determined binding free energies (Table 4) can be contributed to that PMFs were derived from a one-dimensional SMD simulations. Nevertheless, the experimentally best binder [Phe176]pS(174-182) is a second best binder from the PMF calculations."

3. We have revised Fig 1 and its legend:

"The structure is the largest cluster representative of the MD simulation of the complex.  Monomer structure of 14-3-3ε is presented in gray ribbon, and the peptide backbone is shown in red tube. The pSer178 residue of the peptide is shown in green stick, and Lys50, Arg130 and Tyr131 residues of 14-3-3ε that interact with pSer178 are shown in colored sticks."
Also in response to another reviewer, we have created Figure S6  depicting detailed residue-residue interactions between [Phe176]pS(174-182) and 14-3-3ε (Fig S7A and B). In Fig S7C  all residues of the same peptide in the binding groove of 14-3-3ε is depicted. Amino acid residues are labeled in one letter code.

4. Reference 36 is properly indicated in the second paragraph of section 4.1.3.:

"Convergences of simulations were determined by following the method of Hess [36]."

Reviewer 5 Report

Comments and Suggestions for Authors

This paper addresses a significant challenge in cancer therapy, particularly in the context of cutaneous squamous cell carcinoma (cSCC), by targeting the antiapoptotic activity of 14-3-3ε protein. The rationale behind inhibiting the 14-3-3ε-CDC25A interaction is sound, as it offers a specific molecular target for intervention. The methodological approach adopted, involving in silico design and molecular dynamics simulations, demonstrates a thoughtful and systematic strategy for developing peptide analogs. Shortening the parent pS peptide to 9 amino acid residues and subsequently introducing modifications based on binding motifs of 14-3-3 proteins are logical steps towards optimizing peptide binding affinity. The use of conventional molecular dynamics (MD) and steered MD simulations, along with biophysical methods, adds depth to the study by providing insights into the binding dynamics of the designed peptides. The results presented are informative, indicating a reduction in peptide affinity upon shortening but subsequent rescue of binding through specific amino acid substitutions. This finding highlights the importance of structural considerations in peptide design and optimization. The implications of the study are significant, as the optimized peptides hold promise as candidates for inhibiting 14-3-3ε - CDC25A interactions in cSCC, potentially promoting apoptosis in cancer cells. However, it would be beneficial for future studies to validate the efficacy of these peptides in cellular or animal models of cSCC to corroborate their therapeutic potential.

Overall, this paper contributes valuable insights into the rational design of peptide analogs targeting protein-protein interactions implicated in cancer progression. It underscores the importance of computational approaches in drug discovery and lays the groundwork for further preclinical and clinical investigations in the field of cancer therapy. In conclusion, the paper is well-written, methodologically robust, and offers significant contributions to the field of cancer research, particularly in the development of targeted therapies for cSCC. I have the following specific concerns that need to be addressed before the publication of the manuscript.

·        It is unclear what was the rationale behind shortening the peptide. This should be discussed in the paper.

·        Standard deviation error bars are missing in Figrue 3.

·        Fig. 1 shows the structure of 14-3-3ε – pS(174-182) complex. How/where was this structure derived/taken from? It should be mentioned in the figure legend.

·        The consensus chemical structure of the peptide should be illustrated in order to better understand the description within the manuscript.

·        Secondary Structure of the Peptide Analogs: Authors describe these results qualitatively only. Quantitative interpretation is required here.  The software packages like K2D2 can be used to derive the % of α-helix, β-sheet and random coil the total structure of each peptide.

·        The figure showing the binding of peptide shortened peptide in the 14-3-3ε (at least for the best short peptide) may help reader in visualizing the binding pocket.

Author Response

  1. Rational for peptide shortening:  

    In the first paragraph of the discussion, at line 253 of the original manuscript we inserted the followings:

    “Shortening a peptide generally improves its enzymatic stability since less number of peptide bonds are available for degradation. Furthermore, the eventual removal of the Met residue from the parent peptide (Table 1) improves oxidative stability of the resultant analog.“

  2. Fig 3. standard deviation error bars are missing:
    In the figure the residue contribution to binding is depicted that was determined by  MM/PBSA calculations which is a single trajectory method and standard deviations were not calculated.

  3. Fig 1. is revised and in the legend we inserted the following:
    "The structure is the largest cluster representative of the MD simulation of the complex."

  4. "The consensus chemical structure of the peptide should be illustrated..."  
    We do not understand what the Reviewer means by "consensus chemical structure of the peptide". We have given in tables 1 and 3 the primary structure (sequence) of the peptides. Also, we have given the consensus peptide binding motifs of 14-3-3 proteins in table 2. 

  5. "Secondary structure of the peptide analogs:"
    We followed the reviewer's suggestion and using the CDSSTR method in the DichroWeb server we analyzed the ECD spectra of the peptides and the results are included in the supplementary material as Table S7.
    Also, in line 199 of the original manuscript after "Figure 5" we inserted "and Table S7"

  6. Following the reviewer's suggestion, we have created a detailed figure as Figure S7, depicting detailed residue-residue interactions between [Phe176]pS(174-182) and 14-3-3ε. (Fig S7A and B). In Fig S7C  all residues of the same peptide in the binding groove of 14-3-3ε is depicted. Amino acid residues are labeled in one letter code.
    Also, in line 314 of the original manuscript after 14-3-3ε we inserted "(Figure S6)"

Round 2

Reviewer 4 Report

Comments and Suggestions for Authors

In this revised version the authors have answered my comments. 

However, the sentence:

Nevertheless, the experimentally best binder [Phe176]pS(174-182) is a second best binder from the PMF calculations.

It is not clear because the [Phe176]pS(174-182)  binder does not have the most negative binding free energy in Table 4.

Two small errors:

In Table S1 the word peptides is duplicated.

In Fig S6 (A) I do not see pSer178 in red.

Author Response

  1. "It is not clear because the [Phe176]pS(174-182)  binder does not have the most negative binding free energy in Table 4."
    Indeed, in table 4 the  [Phe176]pS(174-182) doe not have the most negative values, but those values are calculated from the experimental Kd, but overall the peptide is still  in the best binder group. In our sentence we referred  PMF-calculated shortened peptide results.
    Therefore, we have changed the sentence to:
    "Nevertheless, among the shortened peptides,  [Phe176]pS(174-182) experimentally has the second best KD."

  2. "In Table S1 the word peptides is duplicated." 
    We deleted the duplication
  3. "In Fig S6 (A) I do not see pSer178 in red."
    we corrected the figure legend as follows:
    "...gray ribbon and red tube, respectively.Atoms in pSer178 are reperesented as H, white; C, cyan; O, red; P, yellow stick. Phe176 is shown in red sticks."

    Also, in line 327, we replaced the wrong S4 table number with S6.